# Perceptions of Health Needs among Venezuelan Women Crossing the Border in Northern Chile during the COVID-19 Pandemic

**DOI:** 10.3390/ijerph192215175

**Published:** 2022-11-17

**Authors:** Alice Blukacz, Alejandra Carreño Calderon, Alexandra Obach, Báltica Cabieses, Jeniffer Peroncini, Alejandra Oliva

**Affiliations:** 1Programa de Estudios Sociales en Salud, Instituto de Ciencias e Innovación en Medicina, Facultad de Medicina Clínica Alemana, Universidad del Desarrollo, Santiago 7610658, Chile; 2Department of Health Sciences, University of York, York YO10 5DD, UK; 3Subsistema Chile Crece Contigo, Subsecretaria de la Niñez, Ministerio de Desarrollo Social y Familia, Santiago 8320000, Chile

**Keywords:** international migration, women’s health, inequities

## Abstract

In parallel to the COVID-19 pandemic, Chile has experienced a significant influx of international migrants, many of whom are Venezuelan women who have entered the country through unauthorized crossing points. In this context, gender and migration intersect as the social determinants of health, leading to their experiencing a range of adverse events. This poses important challenges in terms of short- and long-term health outcomes, the social determinants of health, and access to healthcare. This study aims at describing Venezuelan women’s perceptions of their health needs as they migrate to Chile via an unauthorized crossing point, with a focus on adverse events throughout the migration cycle, self-reported health needs, and responses. A qualitative case study was carried out with 22 participants in the Antofagasta region of Chile, including Venezuelan migrant women, healthcare professionals, and social workers from the public healthcare system, stakeholders from non-governmental and international organizations, and local government officials. The semi-structured, individual interviews were analyzed thematically. The results show that Venezuelan women face a range of adverse events throughout the migration cycle. The perceived health needs that are reported are sometimes linked to these adverse events or existed prior to migration and were exacerbated throughout the migratory cycle. Addressing these physical and mental health needs is essential for short- and long-term individual and public health; however, despite substantial efforts to ameliorate the situation, persisting gaps in access to care are reported.

## 1. Introduction

Accounting for disparities in health and healthcare at a global level, particularly in low- and middle-income countries (LMICs), calls for a focus on the population groups experiencing the highest degree of social vulnerability, among whom women migrants are high on the list [1]. Although Chile is considered a high-income country, with a GNI per capita of USD 24,020 in 2020 [2], its GINI coefficient of income inequality is 46, placing it among emerging economies [3]. Furthermore, 53% of its households are considered economically vulnerable; the poorest 20% of households represent only 5% of the total earnings [4]. These inequalities, from the perspective of the social determinants of health, unequivocally jeopardize population health. Social vulnerability can affect all aspects of someone’s life, from nutrition throughout their life course to health and safety at work, exposure to violence, and environmental or man-made disasters, as well as early-life education and social protection in old age [5,6,7,8,9]. 

In addition to this wider context of socioeconomic disparities, the health system is fragmented between the public system (Fondo Nacional de Salud (FONASA)), the private system (Instituciones de Salud Previsional (ISAPRE)), and the Armed Forces and Police system. In terms of the provision of care, the institutions covered by FONASA are public healthcare centers managed by the Sistema Nacional de Servicios de Salud (SNSS). The institutions covered by ISAPREs are privately owned healthcare centers or independent healthcare providers, and the Armed Forces and Police system has its own dedicated centers [10]. Given this context of important socioeconomic inequalities and inequities in access to health, Chile is a relevant country to focus on when studying health and healthcare inequities. International migrants who recently arrived in Chile through an unauthorized crossing point, among whom are a growing number of Venezuelan nationals, experience a high degree of social vulnerability [11,12]. 

International migrants are defined by the International Organization for Migration (IOM) as people who move away from their place of usual residence across an international border, either temporarily or permanently, and for a variety of reasons [13]. Chile is home to nearly 1,500,000 international migrants, half of whom are women [14]. Migratory outflux from Venezuela is a phenomenon with important regional repercussions in South America, and it is estimated that 4.1 million Venezuelan nationals live in the continent outside of Venezuela [15]. Venezuelan migration to Chile started growing in 2012, from 5748 residents that year to 83,045 in 2017 and accelerated between 2017 and 2021 to reach nearly 450,000 [11], making the Venezuelan community the largest among international migrants in Chile. In parallel, unauthorized migration grew when the requirement for Venezuelan nationals to apply for a visa before entering Chile was established in 2018; this phenomenon was exacerbated when all borders were closed in 2020 as a prevention measure during the health emergency caused by the COVID-19 pandemic [12]. Unauthorized migration or irregular migration is defined by the IOM as the “movement of persons that takes place outside the laws, regulations, or international agreements governing the entry into or exit from the state of origin, transit or destination” [16]. In Chile, this means being in the country without a valid visa as a result of either entering through an unauthorized crossing point, overstaying as a tourist, or staying in the country without renewing an expired temporary visa [17]. In the first half of 2021, it is estimated that almost 18,000 people entered the country by foot or via informal transportation by means of unauthorized crossing paths in the northern region of the country [18]. In terms of the sociodemographic characteristics of Venezuelan residents in Chile, 50% are women, the majority of whom are between 20 and 35 years old [6]. With regard to access to healthcare, according to data from the 2017 Caracterización Socioeconómica Nacional Survey (National Socioeconomic Characterization Survey, CASEN), which offers the most recent data available, 58.5% of Venezuelan nationals in Chile are covered by the public health system, FONASA, 16.5% by the private system, ISAPRE, and 22.1% reported not being covered [19]. Additionally, in terms of the use of healthcare, the same dataset reveals the underuse of services relative to needs. This dataset pre-dates the pandemic; considering the high number of migrants entering through unauthorized crossing points in 2021, we can expect a higher coverage gap. Finally, the public system presents challenges in terms of adequately covering the needs of its beneficiaries [10,20,21,22], especially as the COVID-19 pandemic overstretched its capacities and delayed the provision of necessary care for chronic diseases and other health needs [23].

This study was conducted in the Antofagasta region, which represents around 16% of the Chilean territory and has 703,746 residents [24], 12.4% of whom are international migrants, which is twice the national rate [25]. Its main sector of economic activity is mining, which represents 57% of all activities, while the main sources of employment are trade and real estate, and the region represents 30% of all of Chile’s exports [26]. Although its poverty rate is lower than the national poverty rate, it grew from 5.1% to 9.3% between 2017 and 2020, and 5.7% of people living in Antofagasta experience overcrowding [25]. The region has historically received international migrants from Peru, Bolivia, and Colombia [27]; however, migrants from Venezuela have become an important part of the migrant population in the region, considering that after crossing into Chile in the northernmost regions of Arica y Parinacota and Tarapacá, they usually transit through, and sometimes settle in, Antofagasta on their way south to the capital city of Santiago [28]. In this context, the current study focuses specifically on Venezuelan migrants in the Antofagasta region. Venezuelan migrants in Chile are a diverse group who experience social vulnerability to different degrees, depending on their migratory and life trajectories, including pre-departure context, year of migration, and reason for migrating [29]. Considering the tightening of migratory regulations specifically targeting Venezuelan nationals and border closures during the COVID-19 pandemic, as well as the worsening of social and economic conditions in Venezuela, recently arrived migrants have been experiencing increased social vulnerability, as reported in previous research conducted in the northern regions of Chile [12,28,30]. This poses important challenges in terms of health outcomes, the social determinants of health, access to healthcare, and the wider humanitarian response to migration fluxes in the context of the COVID-19 pandemic. 

Migration is widely agreed to be a social determinant of health, insofar as the health of migrants may be impacted throughout the cycle of migration; structural mechanisms leading to inequities are reproduced and amplified in the context of human mobility [31]. Gender is also recognized as an important social determinant of health, as established by the World Health Organization (WHO) Commission on the Social Determinants of Health in its 2008 report [32]. Gender and health assessments must go beyond women’s and girls’ physical well-being [33]; they experience specific challenges rooted in structural inequities and gender relations of power that damage their health, such as discrimination, violence, limited access to resources and opportunities, and limited agency over their health [32], all of which also have an impact on their children [34]. 

Furthermore, when gender, migration, and poverty intersect as women migrate, the mechanisms leading to social vulnerability are conflated. Intersectionality becomes relevant in order to analyze these mechanisms and address short- and long-term population health disparities. More specifically, the existing literature, focusing on sexual and reproductive health, shows the importance of gender on the reproduction of inequities among specific groups, such as international migrants. Being a migrant woman implies a double-fold discrimination, intersecting with other factors linked to legal and social status, as well as ethnicity [1]. The evidence shows that in different contexts, migrant women underutilize maternity services [35], are more exposed to the risk of giving birth prematurely, and face barriers to accessing birth control, fertility care, and safe abortion care [36,37,38]. 

In Latin America, from a gender perspective, the existing evidence points to existing inequities in sexual and reproductive health, affecting migrant women in particular [36,37,38]. However, this evidence is focused on maternal health, leaving behind key health issues beyond merely the female reproductive function, such as sexual and mental health. With regard to sexual health, the existing studies focus on HIV among men who have sex with men [39], without addressing other sexually transmitted diseases and the way in which HIV affects women who migrate and are exposed to sexual violence in this process, despite international evidence showing that both migrant women and men are more at risk of sexually transmitted infections (STIs), including HIV and hepatitis B [40]. Regarding mental health, studies using a gender perspective tend to focus on the prevalence of post-partum depression among women migrants; however, there is no evidence as to whether women migrants are at higher risk [41,42]. An important point, however, is that a lack of social support, facing discrimination, and economic adversity are key factors that seem to contribute to affective disorders among women migrants in the context of maternity [43,44,45]. 

In this context, studying disparities in health and healthcare with a specific focus on Venezuelan women migrants in Chile is highly relevant. This article aims to describe Venezuelan women’s perceptions of their health needs as they migrate to Chile through unauthorized crossing points, along with stakeholders’ perceptions of Venezuelan women’s health needs, as well as system responses during the COVID-19 pandemic. 

This study could inform healthcare teams and policymakers in Chile and the Latin America region on the main challenges regarding health outcomes and healthcare access for Venezuelan migrant women, in the context of the current health and migration crisis and beyond.

## 2. Materials and Methods

### 2.1. Paradigm and Study Design 

An exploratory case study was carried out under the qualitative paradigm, which was selected as it allows for an understanding of the meanings and perspectives of the population studied, centered around a specific topic. It also considers the way in which these perspectives shape and are shaped by physical, social, and cultural contexts, delving into the processes that maintain or alter these meanings and perspectives [46]. More specifically, case studies focus on obtaining “an in-depth appreciation of an issue, event or phenomenon of interest, in its natural real-life context” [47]. The case study carried out herein is a collective one, defined by Creswell et al. as addressing a single issue; in this case, the issue is the perceptions of health needs among Venezuelan migrant women living in the Antofagasta region of Chile, after entering the country via unauthorized crossing points, with several cases being used to illustrate it [48].

### 2.2. Participant Selection and Recruitment

The total number of participants was 22: ten Venezuelan migrant women, seven healthcare professionals and social workers from the public healthcare system, four stakeholders from non-governmental and international organizations, and two local government officials. The sample was purposive, as the research team sought to interview foreign-born adult women who had entered Chile through unauthorized crossing points in northern Chile, along with a range of participants from the health and social sectors with experience working with women migrants in the Antofagasta region of Chile. For international migrants, the inclusion criteria were as follows: identifying as a woman, being from Venezuela, having entered Chile through unauthorized crossing points between 2020 and 2021, being registered in one of the shelters managed by Chile Crece Contigo in the Antofagasta region, and being able to hold the interview online. Exclusion criteria were the following: having entered Chile before the 2020 border closure and not being registered with Chile Crece Contigo. Women facing especially complex social or health situations were excluded. For the other actors involved in the survey, the inclusion criteria were working or volunteering with international migrants who entered via unauthorized crossing points in the Antofagasta region from 2020 to 2021 in the health or social sector as a doctor, nurse, nurse assistant, psychologist, occupational therapist, social worker or volunteer. Exclusion criteria were having only worked with international migrants before 2020 or in another region. Data saturation for the main dimensions of analysis of this study was reached at 22 participants.

The recruitment of participants was carried out with the support of the Subsistema Chile Crece Contigo (“Chile Grows with You Subsystem”), which is composed of several programs that are part of the Ministry of Social Development and focus on ensuring integrated social protection for all children between 0 and 9 years old and their families in Chile, including access to healthcare and all related services [49]. Although the Chile Crece Contigo subsystem is not specifically focused on immigrant children, considering the arrival of an increasing number of families with children in the north of Chile since 2020, survey coverage was purposefully extended to them. The regional branch of Chile Crece Contigo reached out to Venezuelan women traveling with children and adolescents and offered them information on accessing support, as well as temporary shelter. In this context, Chile Crece Contigo, through one of its social workers, invited some of its beneficiaries to participate freely and voluntarily in an online or phone interview. First contact was made one-on-one by the social worker, and the main objectives of the study were presented to potential participants. Their contact details were shared only when they explicitly expressed their consent to being contacted by the research team, who called them with further information on the project and to obtain informed consent. It was only once informed consent was freely secured that the interview took place, via a phone or video call.

With regard to the other actors interviewed, the recruitment process was also carried out with the support of Chile Crece Contigo, among other collaborating institutions. Working with the program allowed us to approach the topic of women migrants while including wider aspects that are linked to their families and children.

### 2.3. Setting and Data Collection 

Individual semi-structured interviews were conducted by researchers trained in qualitative research (AC and AO) between June and December 2021, for a duration of approximately 45 min each. The interviews were carried out following an interview guide that was specific to each group and designed by the research team. A summary is presented in Table 1 and the full version is available in the Appendix A. 

The interviews were conducted online, considering restricted mobility and the official recommendations to implement social distancing as a prevention measure against SARS-CoV-2, also known as COVID-19. Conducting qualitative public health research via online platforms has been documented in other studies with successful results [50,51,52]. In most cases, the interviews with migrant women were carried out through a video call to their own mobile phones, although they were offered the possibility of conducting the interview in the Chile Crece Contigo offices so that they would not have to bear the cost of the call. Only one participant asked to take the call for the interview at Chile Crece Contigo, which was carried out in a private room with headphones in order to guarantee her privacy. The other women conducted the interview from their own homes. The healthcare professionals, social workers, health authorities, and non-governmental organization (NGO)/international organization actors performed the interviews from their offices or homes. 

Each participant only participated in one interview. The resulting interviews were recorded for transcription and analysis. 

### 2.4. Data Analysis

The recorded data were transcribed by one of the interviewers (AO) and analyzed thematically by AB and AC. Thematic analysis is a qualitative method that enables thematic patterns to be identified from the collected data [53]. Two researchers (AB and AC) conducted the analysis, based on the following steps: reading the data and margin annotations; identification of preliminary codes (topics emerging from the data); grouping related codes in clusters; creating a codebook with the main codes and subcodes; identifying patterns across interviews. During each of these steps, the researchers compared notes and addressed disagreements until a consensus was reached. All the other authors (AOB, BC, and JP) participated in interpreting the coded data before reaching conclusions.

The analysis was structured around three emerging categories:Adverse events throughout the migration cycle and their impact on health;Perceived health needs;Addressing physical and mental health needs.

Patterns were identified and codes emerged within each category. The information was further organized into subcodes as the analysis was refined. 

### 2.5. Ethics

The study was carried out in accordance with the relevant guidelines and regulations for research involving human beings, including the Declaration of Helsinki, and was approved by the Ethics Committee of the Faculty of Medicine of the Universidad del Desarrollo. All participants filled out an informed consent form before taking part in the interview and could withdraw at any point if they so wished. All data were recorded anonymously and stored on the principal investigator’s computer in a locked file. During and after the interviews, if the participants required it, information on the health system in Chile and how to enroll in the healthcare system, regardless of their migratory status, were given to the participants, as well as information on pro-migrant organizations. A referral mechanism for mental healthcare was also put in place with the Mental Health Center of the Universidad del Desarrollo, in case participants required it; however, none did.

## 3. Results

### 3.1. Description of Participants

We interviewed 10 women from Venezuela, all of whom had entered Chile via unauthorized crossing paths during the years 2020 or 2021, traveling with their children and adolescents. Among them, eight had an irregular migratory status but were in the process of applying for a residence permit at the time of the interview, and two were applying for asylum. Seven of them had been through the *autodenuncia* process, voluntarily declaring to the authorities that they had broken migration laws by entering the countries outside the authorized crossing points, which was defined during the pandemic as a prerequisite to apply for a residence permit, although it usually led to their having to leave the country or being deported [19]. The other three had not completed the *autodenuncia* process, among whom one woman was seeking asylum and another had had their asylum application rejected.

The women interviewed were between 19 and 50 years old and their educational levels varied, with three of them having only incomplete high school studies, five of them having completed their high school studies, and two of them having completed higher education. Most women traveled as part of a broader group of people; some of them traveled with their male partner or brother(s), and one woman traveled alone with her 3-year-old son. 

With regard to the other participants involved in the study, all of them were over 18 years old, had completed higher education, and had experience in working with international migrants before the COVID-19 pandemic.

### 3.2. Coding Tree

Emerging codes and subcodes were organized around the following three main categories of focus of the study (see Figure 1).

### 3.3. Description of Results

#### 3.3.1. Adverse Events throughout the Migration Cycle

Sometimes, international migrants can face adverse events throughout the different phases of migration, starting from their country of origin all the way to settling in a new country [54]. From a social determinants of health perspective, these events may have an impact on their physical and mental health in the short, medium, and long term, undermining health equity. 

Violence

The first important set of adverse events identified among the participants’ stories is related to violence, which was experienced at different points in their migration cycle. First, in their country of origin, a few participants described facing violence or persecution that could be characterized as political, and one of them explained that it directly affected her physical and mental health: “I had buses going into Colombia as well, and the Maduro regime burnt them, because we did not participate in its protests and rallies […] they went and kicked me out of my office, they psychologically and verbally attacked me, they beat me up. Because of the beating, I had back surgery […] so we had to come here, as they say, like criminals, fleeing from the government” (MM4). In more general terms, experiencing violence was a push factor that was observed by one of the stakeholders interviewed.

Second, accounts of violence, which can be characterized as gender-based, emerged throughout the interviews, in pre-departure contexts, and in transit, as well as in Chile. In transit, women, especially those traveling alone, faced sexual harassment or the threat of sexual abuse, especially by people in a position of relative power, such as bus drivers, smugglers, and border officials; this was reported by both migrant women and the stakeholders interviewed. For instance, one woman described experiencing harassment during her journey: “The men who help you, you never know what their intentions are, especially in the desert, what they could do to you, or take you away, like […] this man in Peru did to me, they think that because they are helping you, they gain some kind of right over your body, that you have to pay them with your body, or that we are so vulnerable and they have all the power” (MM3).

More specifically, in the context of settling in Chile, domestic violence was reported to be common by the healthcare professionals, social workers, and NGO actors; however, only one of the women interviewed reported it, saying that her ex-partner refused to let her see her son. We could not determine whether the other women interviewed experienced domestic violence at all, as none brought it up spontaneously and it was not part of the set of questions included in the interview guide. Furthermore, one social worker explained that many migrant women, not exclusively Venezuelan women, who worked in bars ended up engaging in sex work in order to make ends meet: “I have heard many cases, where they tell you, “I got this job offer where they told me I also had to this other thing (paid sex work) and I initially said no but in the end, I said yes because I had no choice, I got offered a job as a bartender but I make more money doing this” (F7). 

Third, Venezuelan women reported having faced abuse, which may be attributed to xenophobia in transit countries and in Chile. While some of the abuse that was described took place in institutional settings, for instance, from government officials when trying to regularize their migratory status or from employers, some also described discrimination from the local population: “It has been difficult, see? So much xenophobia […] because here, there are other people renting, from other countries, from Peru, Chileans, and all day they complain about us” (MM6).

Multiple forms of violence were experienced by the women interviewed or were more widely observed by the other stakeholders at different stages of the migratory cycle. In many cases, these instances of violence are rooted in the intersectionality of gender and migration; although violence was described in the country of origin, it mainly took place during transit or while settling in Chile. 

Material conditions

Precarious material conditions accounted for an important dimension of the vulnerability experienced by Venezuelan women crossing the border to Chile. Many experienced homelessness when crossing Peru and Bolivia, as well as upon arrival in Chile, a phenomenon that healthcare professionals and social workers described as being exacerbated during the pandemic, as the influx of international migrants entering through unauthorized crossing points grew: “Whole families started to arrive. We did not use to see women, it was usually only men, since the pandemic we see whole families, with children, we did not use to see homeless children, now we do” (F4).

Another important aspect of material conditions with a potential impact on health is food, as participants reported limited access to nutritious food in their country of origin as a push factor, as one participant explained: “What made me come here is that, how can I explain? I have a four-year-old child, and to think he could get ill, that he was not eating properly, and even then, if he got sick, how would I buy medicine?” (MM5).

Later in the migratory cycle, during transit, and during their early days in Chile, nutrition also emerged as an important theme, something that may have been exacerbated by gender, as argued by one stakeholder: “Many women, clearly, because of their role as mothers, would sometimes forgo meals in order to feed their children, because most of them have an average of two, three, four children” (F1).

The adverse events linked to material conditions, as reported by the women interviewed and observed by the stakeholders, were noted across the migratory cycle and focused mainly on homelessness and the lack of food. 

Travel conditions

More specifically, travel conditions were a key source of adversity, because of the geographical characteristics of the migration route from Venezuela to Chile, which encompasses rainforests, deserts, and mountain ranges with extreme climate variations. Participants highlighted suffering from the cold weather and altitude sickness: “Around the border, during transit, in Peru, we were used to the climate, and we started bleeding, because of altitude. In Bolivia, crossing the border (to Chile) in Colchane, we had headaches, we started to bleed through the nose and mouth, my daughter fainted two or three times while reaching the border, because of the cold and the altitude, we are not used to it” (MM2).

Most Venezuelan migrants entering Chile through unauthorized crossing points travel by foot from either neighboring countries or from Venezuela, implying that they walked extremely long distances in hostile geographical conditions, sometimes when pregnant or carrying young children, and some reported using overcrowded informal transportation during part of their journey, putting their physical integrity at risk: “The rain stopped, and we all boarded a truck, about 80 of us, and I was around the middle, holding on to the roof, with my child holding on to my legs, and I felt like the truck was falling, it was horrible, I felt like this huge truck crossing the river was going to roll over in the water” (MM5).

Another adverse event specific to the transit stage of the migratory cycle and emerging from the interviews is theft, as many participants had their possessions stolen along the way or were scammed by smugglers, as one participant recounts: “Before we reached the border, we had to sleep in the desert for three nights, because the person we paid took off with our bags, he stole from fifteen people, fifteen adults and seven children, it was absolutely awful” (MM9).

Finally, the topic of death emerged, as the women interviewed reported either having witnessed it or fearing it, due, on the one hand, to the traveling conditions, or, on the other hand, having faced threats as a continuity of persecution, starting in their country of origin or while crossing the borders. One woman described her experience when crossing the border in the following terms: “We were already in Chile, and of course, we knew we were crossing through unauthorized paths, and the police caught us, and they were armed, I thought we would die that day because they told us they had the order to kill […] and I started crying, I begged them, for the children […] they made us go back [to Bolivia]” (MM4). On a similar note, one NGO stakeholder also pointed to the responsibility of the Chilean state in accounting for migrant deaths on the border as a direct result of its restrictive migration policies: “In Chile, there have been indirect murders, right? Eight people have already died on the border” (ONG1). 

The traveling conditions reported by the participants involve different adverse events, ranging from geographical and weather conditions to theft and an omnipresent perception of danger. In that sense, traveling conditions are an important factor to take into account when addressing the health needs of international women migrants, as they may have an impact on both their physical and mental health.

Migratory status

All participants crossed the border to Chile from Bolivia via unauthorized crossing points, and they described the fear of being caught and deported: “Every time the bus slowed down, I thought the police was coming to get me, that I would get deported, that they would [punish me] for being a criminal, for breaking the law, the fear, the agony, that was the worst I have ever been through. Every time the bus stopped, I thought, “They are coming after me, they are coming to get me, they realized [who I am], it was terrifying” (MM3).

Additionally, entering the country through such paths meant that, although they were all willing to do so, they encountered difficulties in regularizing their migratory status, as Chilean law excludes them from applying for a residence permit outside of amnesty programs [17]. The participants described uncertainties regarding the process and prospects of the permit being approved, as well as fear regarding the *autodenuncia*, a phenomenon that was also stressed by the NGO actors. Additional uncertainties around regularizing their migratory status and fears surrounding arbitrary mass deportations that took place in 2020 and 2021, during which period migrants without a criminal record were deported, represented important stressors for the women interviewed as they feared it may happen to them: “What I understand is that they came for them at one or two in the morning and took them away, because in the shelter where I was, they came at two in the morning, took them away and put them on a plane, we thought this was going to happen to us as well, they kicked them and treated them aggressively” (MM4).

Migratory status, in this case, entering through unauthorized crossing points and having irregular status, is particularly important when considering the mental health of international migrants. Uncertainty and fear were expressed by the participants when addressing their migratory status; the other stakeholders interviewed also mentioned it as a key topic of interest. Issues surrounding migratory status arise during transit, as well as in the arrival and settlement phases, and may contribute, on the one hand, to negative health outcomes and, on the other hand, to difficulties in accessing essential services and healthcare. 

#### 3.3.2. Self-Reported Health Needs

Physical health

Throughout the interviews, participants described a varied range of physical health outcomes and healthcare needs; most of the latter were also identified by the social and healthcare workers who were interviewed.

In the previous section, limited access to nutritious food was reported by many participants, starting from the pre-departure phase in Venezuela, throughout their journey, and during the settlement phase in Chile. Malnutrition was identified by the stakeholders interviewed as a major physical health outcome among women in general and specifically in pregnant women, nursing mothers, and children: “On several occasions, we saw women who were six or seven months pregnant but who were barely showing, because they were so malnourished and skinny” (F4).

Another participant reported exacerbated kidney pain and attributed it to a lack of adequate food, water, and toilet facilities on her way to Chile. Importantly, many women described a range of physical symptoms that they connected to stress and the physical manifestations of stress: “I was so stressed out that I could not eat on the way here. So many things happened to me, my stomach hurt, I got my period, it was awful (…) I felt so sick, and I lost a lot of weight, I was emaciated, hollow-eyed” (MM3).

More explicitly, regarding specific healthcare needs, sexual and reproductive health was a prominent topic. Routine checks, birth control, pregnancies, and the prevention, detection, and treatment of sexually transmitted diseases were the most pressing issues: “We have seen many more pregnancies this past couple of years, and here, at least, the reality in this healthcare center is that most pregnancies are from immigrant communities, I think about 70%. So, this is an important need for migrant women, some of them get to Chile almost at the end of their pregnancy” (F1).

Additionally, when asked about their healthcare needs, many women referred to needing medical attention for their children, and the stakeholders interviewed reported having witnessed children in bad physical health as a consequence of the extremely difficult conditions during transit: “A few months ago, during the winter, a group of people crossed through an unauthorized path, and they came here with their things, [having] the intention to stay in Chile […] They got here, to San Pedro, with almost nothing, and there was a baby with them, about six months old, less than a year old, in very bad physical condition, he had to be taken immediately to the hospital” (F3).

Chronic illnesses were another source of healthcare needs, usually diagnosed in their country of origin and exacerbated by lack of treatment and the travel conditions: “[My husband] is stressed and he does not really get any rest, he has been restless. He is chronically ill, he has diabetes, so all of this, it builds up” (MM3).

Finally, COVID-19 was, surprisingly, a less prominent topic among the participants, who seemed overwhelmed by other worries, leaving little space for pandemic anxiety: “When we crossed the border, we did not even think about COVID, we just wanted to arrive.” (MM6). Only one participant reported having been infected with COVID-19 during transit; healthcare professionals and health authorities referred to occasional small clusters while pointing out that international migrants complied with the preventive quarantining upon entering the country.

Self-reported physical health outcomes and needs by the participants and those reported by the stakeholders based on their experience encompass a range of physical symptoms experienced throughout their migratory trajectory, an event that also exacerbated chronic illness, late-stage pregnancy, and other sexual and reproductive health concerns. This is relevant from the perspectives of life course and social determinants of health, whereby health issues and needs may emerge and/or be exacerbated during, or as a result of, migration, with a potential long-term impact on individual and collective health if they remain unaddressed.

Mental health

As anticipated, with the physical manifestations of stress reported by the participants, mental health was an important aspect emerging from their stories, as well as from the observations made by the other participants. It is important to note that mental health is understood herein as referring to mental well-being; the study is not based on formal diagnoses or professional mental health assessment. In that sense, this section focuses on self-reported experiences and perceptions surrounding mental health and identifies potential stressors. 

First of all, the women interviewed expressed feelings of fear and guilt with regard to their migratory experience. Fear was mainly related to the adverse events encountered and described in the previous section, as well as from anticipating the worst possible outcomes for themselves and their family. For instance, one woman described her experience in the following terms: “We faced many difficulties, every border we crossed with the children, we were scared, obviously, because we were with them and we did not know what would happen to us on the way, and stuff did happen, it was very scary, they are very young, but we had to be courageous, strong, and keep going” (MM2).

Guilt, on the other hand, was expressed specifically as a result of being very aware of having broken migration laws while feeling that they had no other choice: “When I boarded the bus in Arica, I got my ticket at six in the morning, and they told me the earliest bus was at nine, and I had not eaten anything, I felt like everyone was looking at me because I was a criminal, you know? I felt like everyone was looking at me, was seeing me. Obviously, I was covered in dust, dirty, disgusting, because I had to walk to cross the border. I felt like everyone was looking, that the police were looking” (MM3). This was also something that was observed by the stakeholders interviewed, as one described: “But then, when that family arrived, they were immediately sent to Calama, the police took them there, so I think, in some way, they feel judged because the police go with them, they feel bad” (F3).

More broadly, these feelings contribute to and exacerbate migratory grief, or the mourning associated with the losses experienced in the migratory process, which was expressed by the participants, thus: “Actually, many of us migrants, really, if you look at it, we need mental healthcare, we need a psychologist, something like that. Because with everything we went through in our country, we get here feeling very low, devastated, going through all sorts of things. And then we get here, and we do not have support, obviously, we need it” (MM3).

From a gender perspective, Venezuelan women migrants suffer from the double burden of migratory grief and the mental load associated with the roles and expectations surrounding being a woman, wife or partner, and mother. The feminine mental load is defined as “the combination of the cognitive labor of family life—the thinking, planning, scheduling and organizing of family members—and the emotional labor associated with this work, including the feelings of caring and being responsible for family members but also the emotional impact of this work” [55]. This issue was reported by the healthcare workers, social workers, and NGO and international organization stakeholders, and a participant expressed the following: “As I told you, I am their rock, if I fall apart, they fall apart. Sometimes they go out first and I stay in and want to cry. I mean, I had a home, my things, leaving my family, my mother… So sometimes I tell them, go ahead, I will catch up with you, and I stay in and cry” (MM4).

In terms of the symptomatology of compounded stress and anxiety, several participants described having suffered intense fits of crying as an immediate reaction to a difficult event or as a way to cope with past events: “I went to the police and when I got there, you know, I had some kind of crisis, I cried and screamed like never before, I asked for forgiveness a thousand times, because my intention was never to break the Chilean law because, as I told you, not all migrants are here to commit crimes, we only come here for a better future for our family” (MM3).

Finally, the other participants, among whom were healthcare workers, social workers, and NGO stakeholders, reported that mental healthcare was an urgent need, considering the adversity faced by Venezuelan migrants throughout their migration cycle: “Well, I have seen the impact on mental health because, as I said at the beginning, they are very sensitive, with a lot of repressed emotions, I can imagine, because of the suffering, the pain, everything that comes with migrating informally” (ONG2).

#### 3.3.3. Addressing Physical and Mental Health Needs 

Health system

From a life course and migratory trajectory perspective, it is important to first look at access to healthcare in the migrants’ country of origin, as well as during transit. Some participants left Venezuela because their health needs could not be addressed: “Why did we decide to emigrate? We were looking for a better quality of life, we needed to find help for our children, but also because I have an illness, I have hypothyroidism and I had not been taking my medication for about six or seven years because I could not afford it. If I used the money to eat, I could not use it to buy medication, and I really let myself go, I stopped taking my medication in order to survive, that is how we lived” (MM3). Others reported not being able to access healthcare in the countries of transit:

Interviewee: “No, because I did not know, I thought it was like there, that I could not go to the doctor.”

Interviewer: “So, all those years you spent between Colombia, Peru, and here, you did not get healthcare?” 

Interviewee: “No.” (MM1)

Once in Chile, both the women interviewed and the healthcare professionals, social workers, and NGO actors report a range of barriers to accessing healthcare. The most prominent barrier is related to discrimination, both institutional and individual. International migrants in Chile have the right to access public healthcare, regardless of migratory status, as dictated by Decree 67 in 2016 [56]. However, this right is not always guaranteed in practice. On the one hand, FONASA, the institution providing public health coverage, put obstacles in the way of granting coverage to irregular migrants: “We just could not get coverage [for] people with an irregular migratory status. I mean, if I ever made the mistake to tick “irregular status” on the form, it would get rejected. So then, we just stopped ticking the box, and they started to introduce time limits” (F2). On the other hand, individual administrative staff or healthcare providers would also deny access to healthcare, based on migratory status, an act of discrimination that many of the participants experienced or reported witnessing: “Our colleagues, I do not know if it happens everywhere, sometimes [they] are not very open-minded, with regard to access to health and they make it more difficult […] They say that ‘everything is too easy for them’, [ask] ‘why should [we] give them everything, like healthcare, so easily’, that ‘they will all come here, if we give them healthcare, they will all come asking for it’, [and ask] ‘Why do they come to other countries to have children?’. What else? ‘They have not been here long enough’ […] In primary healthcare, the process at first is administrative, and the administrative staff sometimes make it harder because they feel like it is unfair” (F2). 

In turn, this often led the participants to fear rejection and mistreatment, avoiding seeking healthcare: “The truth is that about three months ago, I went to a healthcare center and they would not see me, and yes, I have been unwell at times, but what is the point [of going], they will say the same” (MM2). This intersects with the fact that many of them did not know they indeed had the right to access healthcare: “We needed (healthcare) when we got COVID, we knew it was COVID but we did not go, because we did not want to go through the *autodenuncia*, and if we went, obviously we are here illegally, so we just waited it out” (MM8). Finally, for those who considered resorting to healthcare via the private system, the cost was an important barrier.

Despite the important barriers identified, not all participants had difficulties in accessing care; facilitators within the health system were highlighted, such as the existence of programs dedicated to promoting migrants’ access to healthcare in public primary healthcare centers, as well as staff trained to assist migrants in carrying out the necessary administrative processes to register with the public system: “I give out essential information on how to go [and] start the migratory regularization process, and I get them through the process of getting health coverage” (F5).

In a similar way, the healthcare system and local authorities strove to promote COVID-19 prevention among migrant communities, including educational talks, the provision of face masks, and access to quarantine facilities: “We contributed with education and awareness because we did not have the resources to give away face masks, sanitizer, etc. […] At the time, we were not in charge of PCR tests, but we made the necessary arrangements to facilitate access” (F2).

Finally, intersectoral coordination was organized between the health system and other governmental agencies, international organizations, and civil society organizations to address specific issues, including child protection and gender-based violence: “We are implementing emergency shelters for recently arrived migrants and to avoid childhood homelessness, to comply with international laws and frameworks. We also try to coordinate with IOM and the *Servicio Jesuita a Migrantes*” (AU2).

Civil society and other actors

Aside from the healthcare system, other key actors emerged in response to the health and wider social needs of recently arrived Venezuelan migrants. Addressing these social needs is relevant to health from a social determinant perspective. 

First, with regard to the immediate humanitarian response to migratory fluxes on the Chilean border, international organizations and civil society organizations provided food and shelter; one of the participants described benefiting from this provision: “There was a lady who helped us a lot, I do not know whether she was from UNHCR, I think UNHCR was paying her to make us breakfast, lunch, and dinner” (MM4).

Civil society organizations are also instrumental in promoting access to the healthcare system, either by providing information to international migrants regarding their rights and where to seek care or by supervising the process of gaining access to healthcare when it was previously denied: “This woman from FASIC [an NGO] told me that I could access healthcare, including the dentist. That I could just go and would not have to pay for anything”(MM1).

Finally, and more widely, the Venezuelan women interviewed reported that despite experiencing discrimination and xenophobia, some members of the local communities were supportive and sometimes provided food and water, included them in community cookouts, or bought from them: “Yes, some people helped us a lot, even now, I go out to sell cookies and they help me a lot, I do not have any complaint regarding Antofagasta, Chilean people are very nice” (MM4).

In the specific context of the pandemic, *ollas communes* or communal meals were organized by the local population, in order to ensure that fellow members of the community or a neighbor could eat, an act of solidarity that has historically been a part of communities’ response to crises in Chile, and included international migrants: “When the pandemic started, communal meals were organized, and they were very inclusive. It did not matter whether you were foreign, Peruvian, Bolivian, Afro-American, blond, dark-skinned, it did not matter. Everyone got the same meal” (ONG1).

Personal strategies

Considering the important barriers to accessing healthcare when needed, participants reported a range of individual strategies to address their health needs. To address physical healthcare needs, they self-medicated with medication bought in informal settings or with alternative medicine, such as herbal teas: “We had to go back to Colombia, again. We were all good in Ecuador and we went back to Colombia because I got very sick (with COVID) and I thought I was going to die, and I said, “If I am going to die, I would rather die in my homeland, I am close, I need to go back”, so we started the journey back, we slept on the streets in Colombia while I was recovering, and thank God, with herbal tea, I got better and we carried on with the journey”(MM4).

With regard to mental health, participants used a range of coping strategies, for instance, religion: “I just turned to God when I had to cross the stretch controlled by the guerrilla” (MM8). Others expressed turning to resilience in the face of adversity: “We had to be courageous, be strong, and carry on” (MM2).

However, some reported wanting to forget parts of their experience: “I try to erase everything I went through in Peru, because really, they attacked us so much there” (MM10).

The results presented show that Venezuelan women face a range of adverse events throughout the migration cycle, among which violence, precarious material and travel conditions, and difficulties in regularizing their migratory status due to having entered Chile via unauthorized crossing points. The perceived health needs that were reported are sometimes a result of these adverse events, especially with regard to mental health, where episodes of stress and anxious symptomatology are reported by the participants as a consequence of their migratory process or existed prior to migrating, sometimes being the reason for leaving Venezuela. Others report seeing chronic illness being exacerbated due to a lack of access to medicine and medical attention in their country of origin and during transit. Addressing these physical and mental health needs is essential for short- and long-term individual and public health; however, despite important efforts, persisting gaps in access to care are reported.

## 4. Discussion

The 2030 Agenda for Sustainable Development outlines a roadmap for reducing poverty and multidimensional inequalities [57]; guaranteeing health and healthcare equity among population groups experiencing high levels of social vulnerability is key to achieving the Sustainable Development Goals on “good health and well-being”, “gender inequality” and “reduced inequalities”. At a regional level, the Cartagena Declaration on Refugees broadened the restrictive and partial definition of the Geneva Convention of 1951, as it recognizes the situations of generalized violence and internal conflicts as sources of forced migration [58]. Although Chile and other transit countries for Venezuelan migrants on the continent adopted the Declaration, the tendency has been to treat migrants from Venezuela strictly as either economic migrants whose presence is subject to obtaining a work visa or as irregular migrants who might face deportation. The lack of a regionally coordinated response to these fluxes leads to policies focusing on control and deportation. The main result of these responses has been an increase in irregular migratory fluxes and the challenges that they bring, including those identified in this study [59].

This paper presents key qualitative data on the health needs of women migrants from Venezuela in northern Chile, in terms of adverse events experienced during the migration cycle, self-reported health needs and outcomes, and multi-level responses. The study is based on 22 interviews with multiple actors, including the voices of Venezuelan women who migrated to Chile via unauthorized crossing points, and draws a picture of the conditions under which women from Venezuela migrate across the continent to settle, permanently or temporarily, in Chile from a health perspective. 

The first key finding is that women face multiple adverse events that can potentially affect their physical and mental health outcomes at every stage of their migration cycle. These events began in the pre-migration phase, in their country of origin, where they reported facing financial insecurity, malnutrition [60], medication shortages, and political and gender-based violence. Venezuelan women were especially affected by the political and economic crisis in their country, as they had to dedicate more time and effort to providing food for their families in a context of extreme scarcity, working two or more jobs to compensate for hyperinflation [61]. A study focusing on Venezuelan women migrants in Colombia showed how migratory conditions and multidimensional exclusion in settlement countries in South America leave them little choice but to work in the informal job market, leading to job insecurity and excluding them from social protection schemes [62]. Furthermore, women in Latin America have lower education levels than their male counterparts, and a significant proportion of their income is used to care for children or other family members [63]. These factors contributed to their decision to leave Venezuela and can determine their health needs, not only during transit, when most participants reported not being able to access care when needed, but also in the immediate arrival phase and in the longer term when settling in Chile. 

Violence also takes a prominent place among adverse events in the migratory cycle, as reflected in some of the participants’ accounts. Besides the political violence experienced in Venezuela, the physical and emotional violence that is reported is often gender-based, which can be defined as “violence directed against a person because of that person’s gender, or violence that affects persons of a particular gender disproportionately” [64], and is experienced sometimes within families or is exerted by other actors during transit or after settlement in Chile. The existing literature [65,66,67] reports a global feminization of migration fluxes, where women travel alone or with children and adolescents, a situation exacerbated during humanitarian crises, when women, despite experiencing a higher degree of economic vulnerability and less decision-power, undertake perilous migration processes similar to those presented in this study. The violence experienced in transit is intimately related to gender, insofar as it is exerted by opportunists, traffickers, or border authorities, who have the power to facilitate or negatively interfere with transit and border-crossing and, thus, consider that women owe them sexual favors in exchange for their help [66,67,68]. This leads to women facing a permanent threat of sexual assault during transit [69,70]. Transactional sex is considered one of the most frequent threats to women in the migratory process and is seldom reported and prosecuted [71,72]. Crossing the border via unauthorized crossing points also involves experiencing the institutional violence of restrictive migratory policies, which often leads to preventable deaths, as was pointed out by one of the interviewees.

However, gender-based violence is not only experienced during transit [73,74]. A recent study conducted with Venezuelan women living in shelters on the northwestern Brazilian border showed that offenders may be intimate partners, relatives, acquaintances, or members of the military and police forces and that most women living in such conditions reported having suffered violence. This has consequences on their physical and mental health, especially with regard to their sexual and reproductive health (pregnancies, exposure to sexually transmitted diseases, and HIV), along with anxiety and depression [75]. Acknowledging these situations as violent and de-normalizing them as part of their daily life is challenging, as has been reported in different studies focused on Latin American women migrants, where the topic is only partially mentioned by the women interviewed while it is emphasized by the other actors, something that is also observed in the present study as well. Economic difficulties, cultural expectations, and migratory status represent important obstacles for women to acknowledge and report situations of violence [68,76]. 

The second key finding is that women face specific health outcomes and healthcare needs that are related to “being a woman”. With regard to specific adverse events, gender and patriarchal structures are manifested more insidiously through the mental burden that women reported experiencing when migrating with their families, leading to mental health needs that must be addressed from a gender-based perspective. As previously mentioned, the mental health of international migrants and refugees has been extensively researched on a global level; there is a consensus with regard to the impact of the adverse events that can occur during the pre-migration, transit, and settlement phases, seen in specific symptoms such as stress, anxiety, insomnia, fatigue, aches and pains, palpitations, dizziness, and somatization disorders [77]. Although there is no evidence that the migrant population presents a higher prevalence of psychiatric diagnoses than the local population, there is evidence that gender can have an impact on the mental health of migrants and on the way in which symptomatology is linked to different experiences, depending on gender [78]. A review carried out in 2020 [79] found that studies focused on the mental health of migrants included gender as a variable, showing that among women, anxiety and the symptoms of depression were linked to having left behind children in their country of origin [80], receiving death threats and experiencing violence in their country of origin and during transit [81], and having experienced stress and anxiety while pregnant, during birth, and post-pregnancy [78]. Similarly, our study shows the consequences that migratory processes can have on the mental health of the women interviewed, and the need to design strategies for their protection and social integration in transit and arrival countries, as established by WHO guidance [82].

Women also reported needing access to sexual and reproductive healthcare, while reporting very limited access during transit and barriers to access while in Chile. Although women’s health should not be reduced merely to maternal health, it is an important dimension considering the fact that many women migrate while pregnant, as reported mainly by the stakeholders interviewed, who have seen an increase in pregnancies among recently arrived women migrants, a phenomenon also reported in the existing literature [83,84]. Another study conducted in Brazil, with Venezuelan women facing similar conditions to the ones included in our study [36], showed the consequences of not approaching the promotion of sexual and reproductive health from a human rights perspective, in terms of the maternal health of women migrants facing migratory irregularity. The study shows results similar to ours, wherein pregnant women migrants are less well-informed of their right to access healthcare and on how the healthcare system works, have limited access to timely prenatal check-ups, have fewer postnatal check-ups, and have limited access to birth control than women who are not immigrants [36]. Additionally, another study focused on the health outcomes of newborns within Venezuelan families in Colombia found that the babies are underweight, presenting a lower APGAR score and affiliation gaps to the healthcare system than children born to Colombian mothers [85,86]. Guaranteeing the right to sexual and reproductive health and to pre- and postnatal care is an essential part of guaranteeing human rights. Despite Chile’s achievements in terms of improving maternal and child health, our results show that this can be reversed, as health inequities are growing among populations experiencing social vulnerability.

The third key finding focuses on the response to adverse events and healthcare needs. While participants highlight important efforts from the healthcare system to facilitate access to healthcare for international migrants, among whom are Venezuelan migrant women, these efforts are undermined by persisting barriers. Although Decree 67 guarantees access to public healthcare regardless of migratory status, barriers related to additional administrative requirements, active discrimination, and a lack of knowledge impose important limitations on achieving this right. This finding is consistent with the existing evidence on access to healthcare among international migrants in Chile, produced before and during the COVID-19 pandemic, suggesting that the situation has not improved in the last few years and may have worsened during the pandemic [28,30,87,88,89]. Additionally, participants reported facing important barriers to accessing healthcare while transiting through other countries, wherein access to healthcare for international migrants, especially those with an irregular migratory status, is not guaranteed [90].

Barriers to accessing formal healthcare often mean that migrants must rely on individual strategies to respond to their healthcare needs, such as self-medication, as reported in other contexts [91]. In Chile, this is mitigated by initiatives to include migration specialists among the social workers in public healthcare centers; however, these instances are local and do not extend to all centers, and civil society organizations played an important role in facilitating access to care. With regard to responding to wider psychosocial and material needs, the work of civil society organizations and international organizations was highlighted, either in coordination with the health sector or with other government institutions. In that sense, although the health system must be at the center of the response to health needs and must strive to eliminate barriers to effective access to care, it is important to consider and include other actors and sectors, from government institutions to civil society organizations, from a multi-sectoral perspective.

These findings show that as gender, migration, and social vulnerability intersect, women migrants from Venezuela who enter Chile through unauthorized crossing points face health and healthcare inequities, due not only to the adverse events experienced throughout their migratory cycle but also to barriers to accessing healthcare, stemming from factors linked to being an international migrant (Figure 2). This calls for an approach to the health of women migrants that considers both migration and gender as determinants of physical and mental health, one that is based on the principles of ethical and cross-cultural care, as well as human rights.

However, this issue cannot and must not be addressed by Chile alone. Considering the fact that adverse events occur throughout the whole cycle of migration, which spans across a continent, a regional approach must be taken. Aside from advocating for improved living conditions in Venezuela, countries in the Latin American region must improve access to health for international migrants and provide a holistic response to forced migration fluxes. 

Strengths and limitations

To the best of our knowledge, the study is unique in Chile, as it integrates gender and health into the analysis of the Venezuelan migratory crisis. Additionally, the study was conducted with a hard-to-reach population that has been made particularly invisible during the pandemic and that is seldom included in scientific research. This study was the result of a collaboration between the academic sector, government entities, and NGOs. It addresses a research gap identified by the social and health sectors with regard to the systematization of the current humanitarian and migratory crisis in northern Chile, in order to create recommendations for future crises.

The main limitation of this study is that the sample size is small and was obtained through convenience sampling, meaning that its conclusions may not be generalizable. As a case study under a qualitative paradigm, this study focuses on the perceptions of different actors and does not aim to include all the dimensions of a complex, multi-layered topic such as irregular migration and its impact on health. As this is an exploratory study, our findings call for future in-depth studies focused on the physical and mental health of women migrants.

Although conducting the interviews remotely via video calls allowed us to gather data in the context of a pandemic and according to social-distancing recommendations, this limited the depth of the study and hindered our fully developing a relationship based on trust between the research team and the participants. Connectivity issues and difficulties in guaranteeing that the participants could hold the interview undisturbed sometimes interrupted the flow of conversation during the interviews. The fact that the women interviewed were mothers and working women, as well as the economic difficulties that most of them were facing, made it impossible to extend the interviews beyond what was stated in the informed consent form. Future research should seek to overcome such limitations by conducting face-to-face onsite interviews or using other qualitative research techniques and conducting a larger number of interviews.

## 5. Conclusions

Our study highlights the experiences of women migrating from Venezuela to Chile via unauthorized crossing points, with a focus on their physical and mental health. Adverse events, such as limited access to food, homelessness, gender-based violence, and political or xenophobic violence are present throughout the migration cycle and have immediate implications for health status, as well as a potential long-term impact that needs to be addressed, not only upon arrival in Chile but also by strengthening regional and intersectoral cooperation, to ensure safe migration processes in South America. Recognizing and addressing these compounded needs and the wider framework of migration as a social determinant of health is key to promoting health and healthcare equity for Venezuelan migrant women in Chile and, more broadly, population health. Nonetheless, our results regarding the responses to the physical and mental health of women migrants in Chile highlight important institutional gaps and persisting barriers to access formal healthcare that are only partially filled by other actors and individual strategies. Additionally, efforts from civil societies have addressed wider social and sanitary needs, such as food, shelter, and basic hygiene. Ensuring health and healthcare equity in Chile and across the region in the long term calls for urgent measures to address migration and gender as intersecting social determinants of health and to mitigate their negative effects on physical and mental health. 

## Figures and Tables

**Figure 1 ijerph-19-15175-f001:**
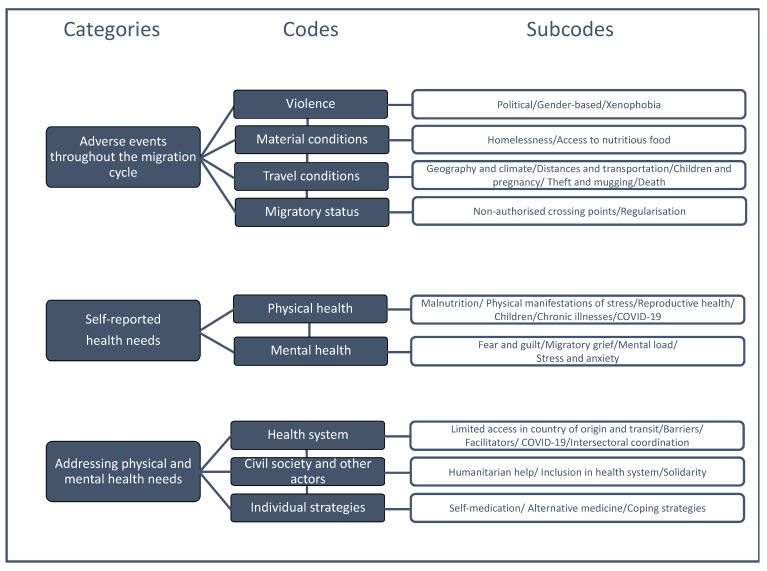
The study’s coding tree.

**Figure 2 ijerph-19-15175-f002:**
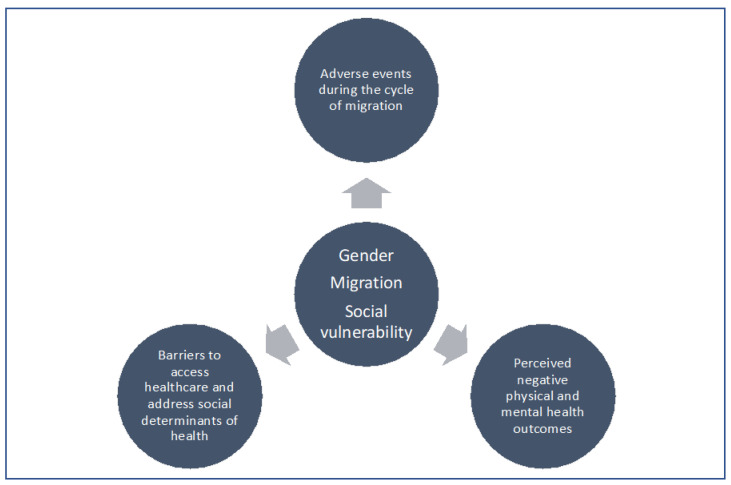
Dimensions of health inequities in the context of intersecting gender, migration, and social vulnerability.

**Table 1 ijerph-19-15175-t001:** Summary of interview guides.

International migrants	Background and reasons to migrate Experiences and health needs in the country of origin Characteristics of the migratory process Experiences and health needs during transit Health system responses in transit countries and in Chile Current living conditions and settling process
Stakeholders	General work experience and experience working with migrants Description of the situation at the border in the region during the pandemic Description of the migrant population using healthcare and social services/other services Main health needs identified Gender dimension of health needs Perception of the response of the health system to the migratory crisis during the pandemic

## Data Availability

The data is available upon reasonable request to the corresponding author.

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
