# Peer review of "Perceptions of Health Needs among Venezuelan Women Crossing the Border in Northern Chile during the COVID-19 Pandemic"

_ijerph, 2022, doi:10.3390/ijerph192215175_

Round 1

Reviewer 1 Report

While I do believe that if conducted properly this study could be very impactful and create changes in the ways that countries handle illegal immigrants, specifically women.  There are some main points that I believe need to be addressed before publication could be considered.

1. The current writing style makes the article read more like a propaganda piece or a news paper article than a well defined scientific study. It is written using a lot of emotive language designed to play on the readers emotions and not on data conclusions. 

2. The article exudes bias in it's writing style and does not have any clear guidelines for data collection.

3. There is little to no objective data presented on this paper, all of the data presented in highly subjective, and there is no evidence of the use of professional mental health evaluation to back up conclusions.

4. I would suggest having each of the immigrant women professionally evaluated by a clinical psychologist/ psychiatrist to determine if the perceptions of the participants are backed up by a clinical diagnosis.

5. Please provide clear inclusion and exclusion criteria as these are currently very vague

6. Please provide a clearer description of the interview process including an appendix detailing the questions asked for each participant/ class of participant. If the participants were not asked the same questions than an argument could be made that the interviewers could have asked biased questions and led the participants using phrasing designed to illicit a desired response. perhaps transcripts of the interviews could be made available.

7. The results are not well formatted with poorly presented and often the conclusions are not clearly backed by the data. There are many sections of the article that make statements without the use of a reference including line 726. These unreferenced statements are emotive and without a clear data basis.

8. This article is defined as dealing with mental health however it only mentions self-reported anxiety and depression. There are many other potential mental health conditions that should be explored.  if the intention is to look solely at anxiety and depression than the title need to be changed.

9. There is also no clear discussion regarding how anxiety and depression or any other mental health conditions are defined and characterised. This creates room for bias in the results.

10. the use of quotes from the interviews has the potential to be used to supplement the data, however this is currently presented as the primary focus of the results and is part of what makes the article read like a news report rather than academic research.

In its current state the article is in my opinion unfit for publication and does not do justice to the importance of the research topic.

Author Response

Please see the attachment 1

Reviewer 2 Report

Dear Authors,

The introduction misses the explanation of the research question of the article. Why did you choose the Venezuelan nationals above others? An explanantion is needed here, as well as a comparison what other nationals are contributing to the migration in the selected region in Chile. Evidence is needed to support the statement that the public healthcare is underfunded and understaffed. At the end of the introduction language barriers and ethnic minority is discussed. Why is this relevant to the article? The national language of both countries is Spanish, and you are not specifiying on ethnic minorities later on. The definition of migrant is missing however. You need to establish who do you view as migrant, and the unauthorized or irregular status must be defined as well. 

The methodology is very weak. In this section the strengths and limitations of the selected case study method must be introduced, together with the typology of the selected case. Is it a deviant case, or a heuristic one? Please see George and Bennett: Case Studies and Theory Development, or Ragin and Becker What is a case? What are the dependent and independent variables of the research?

In the results section there is no visible analysis, in its current form it is just the citation of the interviews. The introduction of the coding tree is not necessary if there is no analysis on the codes. If you keep the coding tree an analysis is needed on how much percentage/etc... of the answers contain the different codes. 

Despite what the title of the article suggested, there is only a minor mention on mental and physical health, and the COVID turned out to be an insignificant contributor if at all. Please consider changign the title of the article or set the focus right in the article. The literature on mental health is weak, if this is the main focus of the article it must be strengthened. In its current state, the article is more about migration than mental health.

The Discussion includes the SDGs, but it is not connected to anything. If a UN level document is mentioned, the article must introduce the regional frameworks as well, at least on migration. Please see the Carthagena Declaration of 1984, more information related to your topic is also can be found in Fiddian-Qasmiyeh (ed.) The Oxford Handbook of Refugee and Forced Migration Studies.

I wish all the best for the Authors.

Reviewer 3 Report

The authors present a very interesting paper on the issue of physical and mental health perceptions of Venezuelan women crossing the border in Northern Chile during the COVID-19 pandemic. The topic itself and its international context constitute the strength of the paper. In turn, the weak point of the study is that the sample size is very small, especially the sample of the Venezuelan migrants, and therefore its conclusions may not be generalized. However, this fact was clearly accented by the authors in the ‘Strengths and limitations’ section. Certainly, this paper is a good starting point for more thorough and broader future research in this field.

I propose to group the contents of the section “3.3. Description of results” into paragraphs. In the present setting of the text, it is legible, but a bit jagged, and therefore unaesthetic.

Authors’ English is correct. The paper requires some editorial/technical amendments but not many linguistic corrections. The research aim was clearly stated. As to the research method, it is a purely qualitative analysis.

The literature presented in the references is relevant and up-to-date and the sources are rich; however, there are some editorial errors or omissions and appropriate complements are required:

– Not all references end uniformly. Normally there should be a full stop at the end, but sometimes it is a semicolon or a full stop preceded by a semicolon.

– Not all articles are described with the number doi, although in fact they have this number, e.g. position 60 in the references, i.e. Anderson, F.M. et al.

– Sometimes a complete bibliographic entry is missing, e.g. in item 57, i.e. Rada, C.P.H, where the numbering of the volume is missing (vol. 15).

Round 2

Reviewer 2 Report

Dear Authors,

With the corrections made the texts is more sound, coherent and comprehensible. The clarifications were much needed to achieve this (added definitions, regional context etc.). 

Further corrections are not necesary in the text, however a minor check on the English language is recommended.

I wish the authors all the best.

Best Regards

Author Response

Many thanks, we have now edited the manuscript to improve the language.